# Intratumoral PD1^+^CD38^+^Tim3^+^ CD8^+^ T Cells in Pre-BCG Tumor Tissues Are Associated with Poor Responsiveness to BCG Immunotherapy in Patients with Non-Muscle Invasive Bladder Cancer

**DOI:** 10.3390/cells12151939

**Published:** 2023-07-26

**Authors:** Debashree Basak, Soumya Mondal, Swadeep Kumar Srivastava, Deborpita Sarkar, Ishita Sarkar, Sukanya Basu, Arpita Bhoumik, Snehanshu Chowdhury, Dilip Kumar Pal, Shilpak Chatterjee

**Affiliations:** 1Division of Cancer Biology and Inflammatory Disorder, IICB-Translational Research Unit of Excellence, CSIR-Indian Institute of Chemical Biology, Kolkata 700032, India; 2Academy of Scientific and Innovative Research (AcSIR), Ghaziabad 201002, India; 3Department of Urology, IPGME&R and SSKM Hospital, Kolkata 700020, India

**Keywords:** non-muscle invasive bladder cancer, BCG immunotherapy, CD8^+^ T cells, biomarker, prognosis

## Abstract

Intravesical immunotherapy with Bacillus Calmette–Guerin (BCG) is a standard of care therapy for non-muscle invasive bladder cancer (NMIBC), which accounts for about 75% of newly diagnosed urothelial cancer. However, given the frequent recurrence and progression, identification of a pre-treatment biomarker capable of predicting responsiveness to BCG in NMIBC is of utmost importance. Herein, using multiparametric flow cytometry, we characterized CD8^+^ T cells from peripheral blood and tumor tissues collected from 27 pre-BCG patients bearing NMIBC to obtain immune correlates of bladder cancer prognosis and responsiveness to BCG therapy. We observed that intratumoral CD8^+^ T cell subsets were highly heterogenous in terms of their differentiation state and exist at different proportions in tumor tissues. Remarkably, among the different CD8^+^ T cell subsets present in the tumor tissues, the frequency of the terminally exhausted-like CD8^+^ T cell subset, marked as PD1^+^CD38^+^Tim3^+^ CD8^+^ T cells, was inversely correlated with a favorable outcome for patients and a responsiveness to BCG therapy. Moreover, we also noted that the intratumoral abundance of the progenitor exhausted-like PD1^+^CD8^+^ T cell subset in pre-BCG NMIBC tumor tissues was indicative of better recurrence-free survival after BCG. Collectively, our study led to the identification of biomarkers that can predict the therapeutic responsiveness of BCG in NMIBC.

## 1. Introduction

Bladder cancers which account for 90–95% of urothelial carcinoma (UC), are considered as the ninth most prevalent cancers worldwide with high morbidity and mortality rates [1,2]. The disease has been staged primarily into two categories, non-muscle invasive bladder cancer (NMIBC) and muscle invasive bladder cancer (MIBC), each having different treatment modalities and disease outcomes [3]. In clinical practice, most of the patients (about 75%) newly diagnosed with bladder cancer have NMIBC, which can further be categorized into four risk groups: low, intermediate, high, and very high-risk group, according to the European Association of Urology (EAU) NMIBC Guidelines Panel, 2021 [4]. Intravesical immunotherapy with Bacillus Calmette–Guerin (BCG) shortly after transurethral resection of the bladder tumor (TURBT) has been the standard of care therapy for patients with NMIBC since its inclusion in clinical urology in 1976 [5,6]. Despite the clinical success, up to 40% of the patients do not respond to the BCG therapy and exhibit frequent recurrence. It has been reported that up to 20–30% of the patients who experience disease recurrence ultimately progress to MIBC, which may not be salvaged with additional therapy [7,8]. Therefore, there is an unmet need to identify biomarkers that can efficiently predict the BCG responsiveness and recurrence in patients with NMIBC.

The prognostic and therapeutic significance of T cells has well been demonstrated in various malignancies including bladder cancer [9]. In fact, multiple studies have shown that T cells play a pivotal role in mediating the therapeutic response of BCG [10,11]. It has been reported that the increased abundance of both CD4^+^ and CD8^+^ T cells in tumor are associated with improved responsiveness to BCG [12,13,14]. However, reports indicate the preponderance of T cells in post-BCG tissues even in the recurrence cases, suggesting an ineffectual T cell response in a fraction of patients [15,16]. Although the predisposition of Th2 CD4^+^ T cells in tumor have been implicated in BCG failure [17], the precise contribution of intratumoral CD8^+^ T cells in influencing the therapeutic efficacy of BCG remains largely elusive.

In the present study, we performed a prospective analysis of intratumoral CD8^+^ T cells from 27 patients bearing NMIBC, based on the cell surface expression of different immune checkpoint molecules and the ability to produce effector cytokines (IFNγ and GzmB). We determined an intratumoral CD8^+^ T cell subset co-expressing PD1, CD38 and Tim3 (PD1^+^CD38^+^Tim3^+^ CD8^+^ T cells) that may serve as a potential biomarker for stratifying patients with different disease outcomes and responsiveness to BCG.

## 2. Materials and Methods

### 2.1. Collection of Tumor Tissues 

Surgically resected tumor tissues (1–2 g in normal saline) along with peripheral blood (2–3 mL in heparin containing tube) were collected from patients with NMIBC before intravesical BCG immunotherapy by medical personnel on obtaining informed consents from patients, with ethical approval from the Institutional Review Board of IPGME&R and SSKM Hospital, Kolkata, India (Ref No. IPGME&R/IEC/2020/624). The tissue samples were collected by transurethral resection of tumor via monopolar electrocautery.

### 2.2. Inclusion and Exclusion Criteria for the Patients

All completely resected NMIBC via TURBT performed under white light cystoscopy were selected for the study. Intravesical BCG immunotherapy was given in intermediate and high risk NMIBC patients. 

Patients who were diagnosed with MIBC were excluded from the study. Additionally, bladder cancer patients with a history of prior recurrence were excluded from the study. 

### 2.3. Single Cell Preparation from Tumor Tissue

To obtain tumor infiltrating lymphocytes (TILs), mechanical dissociation of tissue was done followed by 1 h enzymatic digestion by 1 mg/mL collagenase type IV (stem cell technologies, Vancouver, BC, USA). Following digestion, RPMI-1640 complete media was added to the pellet and filtered through a 70 μm cell strainer (BD Biosciences, San Jose, CA, USA), then centrifuged. Pellet was resuspended in ACK lysis buffer to eliminate red blood cell contamination. Finally, the pellet was dissolved in RPMI-1640 (Thermo Fisher Scientific, Waltham, MA, USA) complete media for further experimentation. 

### 2.4. Isolation of PBMC Peripheral Blood 

To obtain PBMC, collected blood was diluted with PBS in 1:1 ratio. Diluted blood was layered on Ficoll (HiMedia, Nashik, India) and centrifuged at 1500 rpm at 25 °C for 25 min. After this density gradient centrifugation, PBMC layers were collected, washed with PBS twice, and resuspended in complete RPMI-1640 for further use. 

### 2.5. Flow Cytometry-Based Analysis of Immune Cell Subsets

Staining for cell surface markers was performed by incubating cells with antibody at 1:50 dilutions in FACS buffer for 20 min at 4 °C. For evaluation of intracellular cytokines by flow cytometry, isolated TILS and PBMC were restimulated with PMA (100 ng/mL) and Ionomycin (1000 ng/mL) for 4 h in the presence of Golgi inhibitor (Brefeldin A). For intracellular cytokine staining, surface markers were stained before fixation/permeabilization (BD Cytofix/Cytoperm kit, (BD Bioscience, San Jose, CA, USA). Samples were acquired on BD LSRFortessa and analyzed with FlowJo software (BD, OR). Following antibodies were used for staining:

**Table d64e323:** 

Antibodies	Source	Clone#, Cat No#
Anti-Human CD3 BV711	Biolegend	OKT3, 317328
Anti-Human CD4-BUV395	BD Bioscience	RPA-T4, 564724
Anti-Human CD4-Percp cy5.5	Biolegend	RPA-T4, 300530
Anti-Human CD8-A700	Biolgend	HIT8a, 300920
Anti-Human CD8-FITC	Biolegend	HIT8a, 300916
Anti-Human PD1-PECY7	Thermo Fisher Scientific	EBioJ105, 25-2799-42
Anti-Human CD38-APC-CY7	Thermo Fisher Scientific	HIT2, 47-0389-42
Anti-Human TIM3-BV421	Biolegend BD Bioscience	F38-2E2, 345008 7D3, 565562
Anti-Human IFNγ-APC	Invitrogen	4S.B3, 17-7319-82
Anti-Human GZMB-PE	Biolegend Invitrogen	QA16A02, 372208 GB12, MHGB04
LIVE DEAD Fixable Yellow Fluorescent Dye	Invitrogen	L34968A

### 2.6. BCG Administration Timeline

Intravesicular instillation of BCG is the gold standard for patients with intermediate and high-risk non-muscle-invasive bladder cancer (NMIBC). The protocol for BCG treatment is based on the original SWOG trial method which includes a 6-week induction course, followed by a three-weekly maintenance cycle [18]. Briefly, in the present study, based on the histopathological report obtained two weeks post-TURBT, NMIBC patients were stratified as per American Urology Association (AUA) into three NMIBC risk groups: low risk, intermediate or high-risk. Patients either with intermediate or high-risk were treated with BCG therapy following SWOG protocol. To assess recurrence, three-monthly cystoscopy was performed and patients were observed for at least one-and-a-half years throughout their induction and maintenance course of BCG immunotherapy. BCG unresponsive tumors are defined as BCG refractory or a high-grade Ta/T1 BCG recurrence within six months of completion of ‘adequate BCG exposure’ (5/6 doses of initial induction and 2/3 doses of maintenance therapy), or upon developing carcinoma in situ (CIS) within twelve months of adequate BCG exposure.

### 2.7. Statistical Analysis 

Results are shown as mean ± SD and the Mann–Whitney U Test, Pearson correlation, and Spearman correlation analysis were used in this study. The statistical significance was defined as a *p* value < 0.05 (*), *p* value < 0.01 (**), *p* value < 0.005 (***), and *p* value < 0.0001 (****) with two-sided tests, unless otherwise mentioned. Data analyses were performed using the Graphpad Prism 9 software (GraphPad, San Diego, CA, USA). 

## 3. Results

### 3.1. CD8^+^ T Cells in Bladder Tumor Tissues Exhibit Heterogeneity in Their Exhaustion States 

In order to define the phenotype and functionality of CD8^+^ T cells present in the bladder tumor tissues, we performed multiparametric flow cytometry-based analysis of paired samples (tumor tissue and peripheral blood) from 27 patients bearing NMIBC. In our prospective assay cohort, there were total 22 males (~80% of total population) and 5 females (~19% of total population) and the median age of male and female were 55.5 and 45 respectively (patient details are given in Appendix A). First, we determined the frequency of tumor infiltrating T cells in bladder tumor samples, and found a dramatic drop in the frequency of CD3^+^ T cells in the tumor milieu as compared to peripheral blood (Figure 1A). Upon comparing the proportion of intratumoral CD4^+^ and CD8^+^ T cells, we observed a modest decrease in CD8^+^ T cells as compared to CD4^+^ T cells (Figure 1B). As it has been reported that CD8^+^ T cells in the TME can exhibit discrete functional phenotype owing to their cell surface expression of various immune checkpoint molecules [19], we next sought to determine the phenotypic heterogeneity of CD8^+^ T cells present in the bladder TME. Herein, we primarily focused on four important immune checkpoints viz PD1, Tim3, CD38 and Lag3, which have frequently been reported to be expressed on tumor infiltrating T cells [19,20,21]. We observed that as compared to PBMC (peripheral blood mononuclear cells), CD8^+^ T cells at the tumor site exhibited dramatic increase in the expression of PD1 (Figure 1C), CD38 (Figure 1D), and Tim3 (Figure 1E), while Lag3 expression (Figure 1F) was comparable. The data indicates that CD8^+^ T cells present at the bladder tumor site are carrying the markers of dysfunctional T cells. Furthermore, to confirm the dysfunctionality, the overall capability of CD8^+^ T cells to produce IFNγ and GzmB were assessed following re-stimulation in vitro. We noted that both IFNγ and GzmB production were drastically reduced in CD8^+^ T cells from tumor site as compared to their PBMC counterpart (Figure 1G,H), strongly suggesting the functional impairment of T cells at the bladder tumor site. 

It has been reported that exhausted CD8^+^ T cells in TME are highly heterogenous, comprising of various subsets with varied therapeutic responsiveness and prognostic significance [19,22]. Based on the cell-surface expression patterns of three important inhibitory receptors (PD1, CD38, and Tim3), we could demarcate seven different CD8^+^ T cell subsets at the bladder tumor site (Figure 1I). Interestingly, we observed that in addition to single immune checkpoint marker positive CD8^+^ T cell subsets (PD1^+^ T cells and Tim3^+^ T cells), CD8^+^ T cell subsets co-expressing two (Tim3^+^PD1^+^ T cells, CD38^+^Tim3^+^ T cells, and CD38^+^PD1^+^ T cells) or multiple immune checkpoint markers (PD1^+^CD38^+^Tim3^+^ T cells) were also abundant, albeit at different proportions, in bladder tumor tissues (Figure 1J). These data together suggest that different subsets of exhausted CD8^+^ T cells can exist at different proportion at the bladder tumor site and their relative abundance could determine the disease outcome.

### 3.2. Intratumoral PD1^+^CD38^+^Tim3^+^ CD8^+^ T Cell Subset Is Positively Associated with Increased Tumor Burden in Patients with NMIBC 

Given the fact that NMIBC patients display considerable diversity in terms of disease pathology, next, we explored the association of intratumoral CD8^+^ T cell subsets with tumor stage, multifocality and size. We found that tumor stage and tumor size barely had any significant association with any of the intratumoral CD8^+^ T cell subsets analyzed (Figure 2A,B). However, interestingly, a positive correlation was observed with tumor multifocality and the enrichment of PD1^+^CD38^+^Tim3^+^ CD8^+^ T cells (Pearson r = 0.5933, *p* = 0.0011 and Spearman r = 0.5257, *p* = 0.0049) at the tumor site in NMIBC (Figure 2C). Conversely, a negative correlation between tumor multifocality and intratumoral abundance of PD1^+^ CD8^+^ T cells (Pearson r = −0.4266, *p* = 0.0265 and Spearman r = −0.4573, *p* = 0.0165) was observed in NMIBC (Figure 2C). These data indicated the functional diversity of intratumoral CD8^+^ T cell subsets with disparate proclivity to anti-tumor response. 

### 3.3. Intratumoral PD1^+^CD38^+^Tim3^+^ CD8^+^ T Cells Could Predict Disease Recurrence in Bladder Cancer Patients 

Next, the ability of intratumoral PD1^+^CD38^+^Tim3^+^ CD8^+^ T cells to predict the outcome of BCG therapy in bladder cancer was explored. Surgically resected pre-BCG tumor tissue from NMIBC patients, who either had BCG non-responsive tumor or responded with no recurrence with BCG during the study period, was analyzed. Upon comparing the average frequency of different intratumoral CD8^+^ T cell subsets, we noted that patients with BCG non-responsive tumors had ~3 folds higher expansion in the frequency of intratumoral PD1^+^CD38^+^Tim3^+^ CD8^+^ T cells, while the frequencies of other subsets either remained unaltered or was modestly affected (Figure 3A). These observations called for further exploring the density of intratumoral PD1^+^CD38^+^Tim3^+^ CD8^+^ T cells with BCG responsiveness. To this end, we first determined the correlation between EORTC recurrence-risk and frequency of different intratumoral CD8^+^ T cell subsets. Our analysis revealed that while the frequency of PD1 single positive CD8^+^ T cells (PD1^+^ CD8^+^ T cells) was inversely correlated with EORTC recurrence-risk (Pearson r = −0.3853, *p*= 0.0472 and Spearman r = −0.3438, *p* = 0.0791), the abundance of PD1^+^CD38^+^Tim3^+^ CD8^+^ T cells were significantly (Pearson r = 0.5932, *p* = 0.0011 and Spearman r = 0.5524, *p* = 0.0028) associated with EORTC recurrence-risk (Figure 3B). 

The above observation further prompted us to examine if there were any differential enrichment of PD1^+^CD38^+^Tim3^+^ CD8^+^ T cells at the pre-BCG tumor site of responder vs non-responder patients. Indeed, we also found that the density of PD1^+^CD38^+^Tim3^+^ CD8^+^ T cells were significantly higher at the pre-BCG tumor tissues of BCG unresponsive tumors as compared to those who had no recurrence with BCG during the study period (Figure 3C). However, abundance of other intratumoral CD8^+^ T cell subsets were comparable between responders and non-responders. These data together suggest that the proportion of PD1^+^CD38^+^Tim3^+^ CD8^+^ T cells present at the tumor site positively associated with disease severity. Most importantly, intratumoral abundance of PD1^+^CD38^+^Tim3^+^ CD8^+^ T cells appear to be associated with the negative outcome (recurrence) of patients after BCG treatment. 

## 4. Discussion

The abundance of CD8^+^ T cells in the tumor microenvironment (TME) has long been considered as a positive prognostic marker for various human malignancies [23]. However, recent studies have demonstrated the existence of heterogeneous population of intratumoral CD8^+^ T cells, which depending on their state of differentiation determine the therapeutic responsiveness and overall disease outcome in patients with malignancies [19,22]. In the present study we aimed to identify the immune correlates of bladder cancer prognosis and responsiveness to BCG therapy in NMIBC patients. Our data demonstrated that intratumoral abundance of PD1^+^CD38^+^Tim3^+^ CD8^+^ T cells was positively correlated with the progression of the disease and highly enriched in NMIBC patients with high tumor burden. Moreover, the current investigation identified intratumoral PD1^+^CD38^+^Tim3^+^ CD8^+^ T cells as a predictive biomarker for recurrence of the disease and have an inverse correlation with the responsiveness to BCG therapy in NMIBC patients. 

T cells exhaustion, which is defined by the loss of polyfunctionality, self-renewal capacity, proliferation and increased susceptibility to apoptosis can be phenotypically characterized by the sustained cell-surface expression of various co-inhibitory receptors including PD1, Lag3, Tim3, CD38, TIGIT etc. [24,25]. It is apparent from various studies that exhausted CD8^+^ T cells in TME are extremely heterogenous and relative abundance of various exhausted subsets impel the therapeutic responsiveness and prognosis of the disease [19,22,26]. In fact, recent study primarily focusing on the expression of TIGIT and PD1 demonstrated that TIGIT/PD1-based stratification could serve as a prognostic and therapeutic predictor for patients with MIBC [27]. Another study aiming to unravel the immune contexture of NMIBC tumor tissues of pre- and post-BCG inferred the profound influence of intratumoral T cells in predicting BCG responsiveness [14]. Very interestingly, the study demonstrated that while high density of CD8^+^PD1^+^ T cells in post-BCG tissues correlated with BCG failure, increased abundance of CD8^+^PD1^+^ T cells in pre-BCG tumor tissues was found to be associated with BCG responsiveness [14]. The data is in accordance with our observation that intratumoral PD1^+^CD8^+^ T cells exerted an inverse correlation with the adverse outcome of NMIBC patients. As reported in case of viral infection models, we also argued that PD1^+^CD8^+^ T cells at the tumor site represented the tumor epitope reactive immediate effector T cells that were capable of eliciting anti-tumor response [28,29]. In contrast to PD1^+^CD8^+^ T cells, intratumoral CD8^+^ T cell subset expressing Tim3 only (Tim3^+^CD8^+^ T cells) were found to be associated with high tumor multifocality. However, we could not observe any overt enrichment of this population at the tumor site when compared between patients with no recurrence with BCG vs. BCG non-responsive tumors. We postulate that Tim3^+^CD8^+^ T cells possess the features of early exhausted T cells, which can partially retain their effector function and respond to immunotherapy. Nevertheless, comprehensive characterization is needed to further ascertain the functional relevance of this population at the tumor site. 

In the present study, we interestingly observed that intratumoral CD8^+^ T cell subset (PD1^+^CD38^+^Tim3^+^ CD8^+^ T cells) co-expressing multiple co-inhibitory receptors were associated with adverse prognosis and therapeutic failure to BCG in NMIBC. Recently, it has been shown that concomitant expression of multiple co-inhibitory receptors on T cells is a characteristic feature of terminally exhausted T cells which are short-lived, functionally impaired and unresponsive to immune checkpoint blockade (ICB) therapy [21,24]. Mechanistic studies demonstrated a distinctive transcriptional and epigenetic state of terminally exhausted T cells underpinning their inability to mount anti-tumor response at the tumor site [19,21,30,31]. Since both CD4 and CD8 T cells are believed to play a crucial role in eliciting BCG-mediated anti-tumor response, we argued that increased frequency of intratumoral PD1^+^CD38^+^Tim3^+^ CD8^+^ T cells which phenotypically resemble the terminally exhausted T cells and presumably are short-lived, failed to elicit anti-tumor response triggered by BCG in NMIBC. Although, our study focuses on understanding the immune corelates of BCG responsiveness, we propose that the enrichment of PD1^+^CD38^+^Tim3^+^ CD8^+^ T cells at the NMIBC tumor site is not solely related to BCG response, rather it stratifies patients with a high recurrence risk already at the baseline. However, the intrinsic pathway shaping the dysfunctional state of PD1^+^CD38^+^Tim3^+^ CD8^+^ T cells needs further investigation. Moreover, it has also been reported that high intratumoral abundance of terminally exhausted-like CD8^+^ T cells indicated a tumor-promoting microenvironment owing to their positive association with various pro-tumorigenic immune subsets including Th2 cells, Treg, TAM and MDSC [27]. Therefore, it is also possible that high tumor burden and predisposition to BCG failure associated with increased frequency of intratumoral PD1^+^CD38^+^Tim3^+^ CD8^+^ T cells could be in part mediated by pro-tumorigenic immune subsets present at the tumor site. Further studies with a large prospective cohort are necessary to elucidate the complex relationship among these subsets in order to shape up the therapeutic responsiveness to BCG in NMIBC. 

In summary, our present study identified an intratumoral terminally exhausted-like PD1^+^CD38^+^Tim3^+^ CD8^+^ T cell subset that could offer a pre-treatment biomarker capable of predicting the outcome of BCG therapy in NMIBC. Furthermore, PD1^+^CD38^+^Tim3^+^ CD8^+^ T cell abundance was also found to be inversely correlated with the favorable outcome of the patients with NMIBC. The findings would help in stratification of NMIBC patients who might be benefited from BCG therapy or in need of additional intervention before BCG instillation. 

## 5. Limitations

We acknowledge that the study has few limitations. Two major limitations are: (1) Patient cohort of our study is small, and (2) time duration of studying recurrence could have been extended.

## Figures and Tables

**Figure 1 cells-12-01939-f001:**
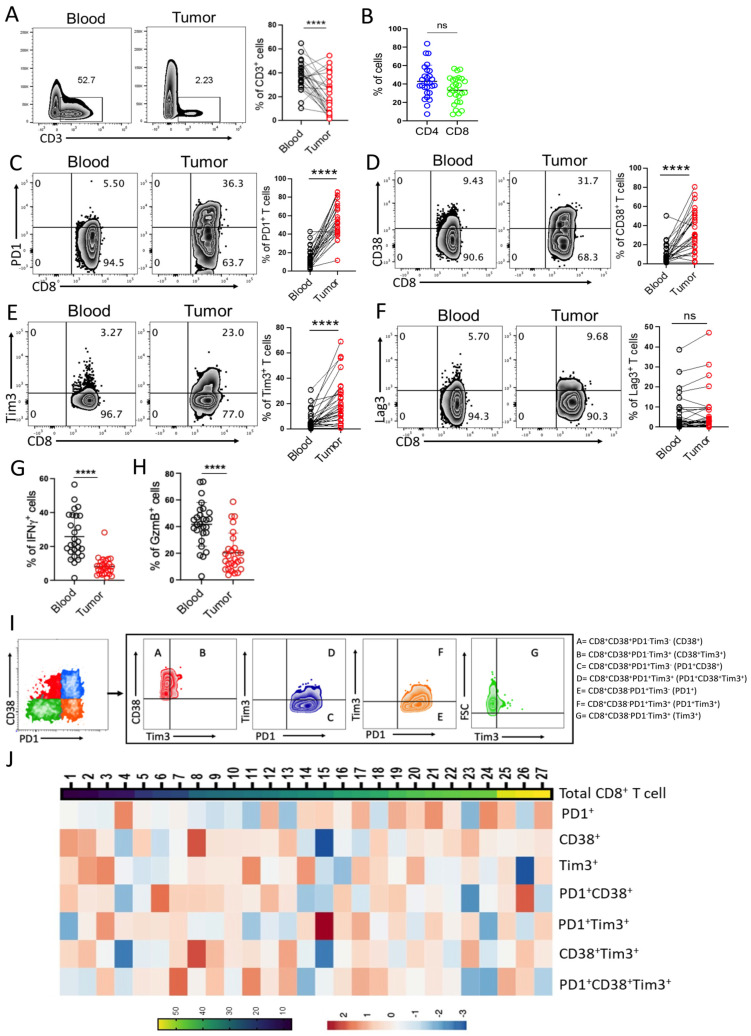
Heterogeneity of CD8^+^ T cells in Bladder Tumor tissue. (**A**,**B**) Flow cytometry-based analysis of: (**A**) frequency of CD3^+^ T cells in peripheral blood and tumor tissues, and (**B**) frequency of intratumoral CD4^+^ and CD8^+^ T cells. Scatter plots represent the cumulative data from 27 NMIBC patient samples. (**C**–**F**) cell surface expression of (**C**) PD1, (**D**) CD38, (**E**) Tim3, and (**F**) Lag3 on CD8^+^ T cells. Adjacent plots represent the cumulative data from 27 NMIBC patient samples. (**G**,**H**) Scatter plots represent the percentage of CD8^+^ T cells positive for intracellular cytokine (**G**) IFNγ, and (**H**) GzmB. (**I**) Gating strategy based on expression of different inhibitory immune checkpoint molecules on CD8^+^ T cells. (**J**) Heat map of CD8^+^ T cell subsets co-expressing various immune checkpoint molecules. **** *p* < 0.0001. The Mann–Whitney U Test was applied to evaluate statistical significance between the groups.

**Figure 2 cells-12-01939-f002:**
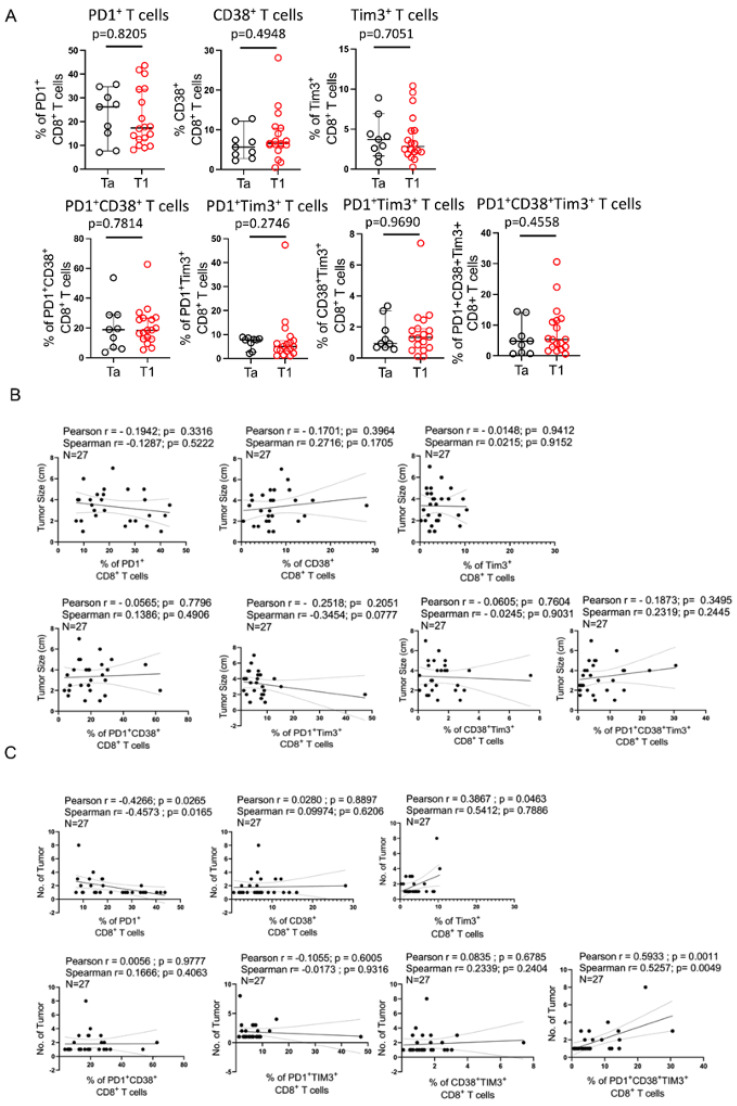
Correlation between Intratumoral CD8^+^ T cell subsets and tumor burden in NMIBC. (**A**) Scatter plots represent the abundance of various subsets of intratumoral CD8^+^ T cells in NMIBC patients with either Ta or T1 tumor stage. (**B**,**C**) Pearson correlation and Spearman correlation between different subsets of intratumoral CD8^+^ T cells with either (**B**) tumor size, or (**C**) tumor multifocality. (**A**–**C**) Analysis done with 27 NMIBC patient samples.

**Figure 3 cells-12-01939-f003:**
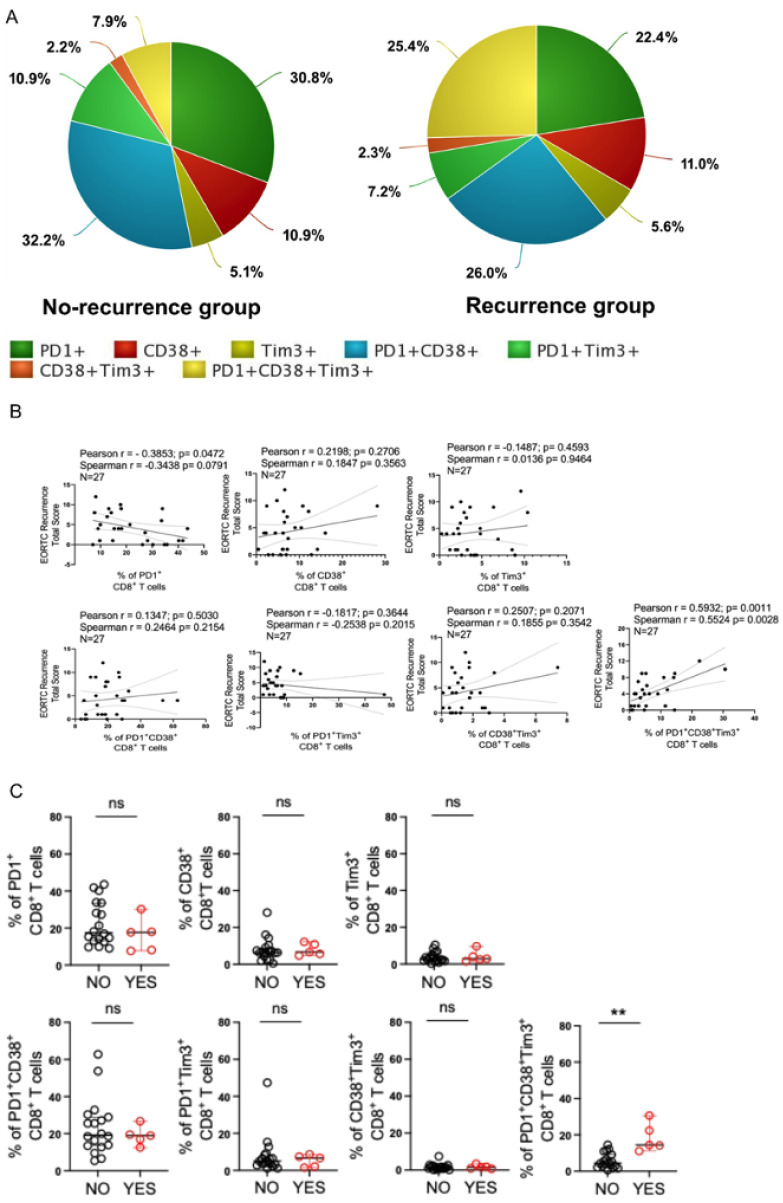
Intratumoral PD1^+^CD38^+^Tim3^+^ CD8^+^ T cells in pre-BCG tumor tissues predict the post-BCG recurrence in NMIBC. (**A**) Pie chart representation of average frequency of 7 different subsets of intratumoral CD8^+^ T cells in NMIBC patients with either BCG responsive tumor or BCG non-responsive tumor. (**B**) Pearson correlation and Spearman correlation analysis between EORTC recurrence-risk and frequency of different intratumoral CD8^+^ T cell subsets. (**C**) Scatter plots represents frequency of various intratumoral CD8^+^ T cell subsets obtained from pre-BCG tumor tissue of NMIBC patients who eventually received BCG and either responded or recurred. Data from 27 NMIBC patients are presented. ** *p* < 0.01. Mann-Whitney U Test was applied to evaluate statistical significance between the groups.

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
