# Peer review of "Intratumoral PD1+CD38+Tim3+ CD8+ T Cells in Pre-BCG Tumor Tissues Are Associated with Poor Responsiveness to BCG Immunotherapy in Patients with Non-Muscle Invasive Bladder Cancer"

_cells, 2023, doi:10.3390/cells12151939_

Round 1
Reviewer 1 Report
In this study, Busak et al. evaluated the expression of PD1, Tim3, CD38, IFNg and GzmB on CD8 T cells in PBMC and tumor tissues of 27 patients NMIBC patients before BCG therapy abd correlated these data to clinical outcome.
Major point:
Their main finding (and it is the article title) was that frequency of PD1+CD38+Tim3+ cells among intratumoral CD8 T cells before BCG therapy was associated to subsequent tumor recurrence. Indeed, this is of great interest to identify immune correlate of BCG failure before therapy (rather than during therapy, or at the time of relapse). However, data do not provide sufficient evidence to support the conclusion.
The % of PD1+CD38+Tim3+ cells among intratumoral CD8 T cells correlated to the number of tumors (Fig2C), to the EORTC recurrence score (Fig3B) and to post-BCG tumor recurrence (Fig3A and C). However, these observations show substantial weaknesses:
- Correlations were tested using a Pearson test. This would need to be validated using a non parametric (Spearman) test since 2 patients with much higher % PD1+CD38+Tim3+ (vs. the other 25 patients) have also a high EORTC score, which may drive the Pearson correlation (while the correlation might be not significant when assessing it among the other 25 patients). Such bias will not occur in a non-parametric correlation test, that is why I think this would be important to know the p-value from the Spearman test.
- Besides, the most convincing would be to compare patients with or without bona fide subsequent recurrence, which is shown in Fig3C. However, the number of patients appear too low to be convincing with only 5 patients with recurrence. Is it possible to test the comparison when recurrence is evaluated at 1 year (for example) rather than 6 months (there might be more patients in the recurrence group in this case). Also, it is unclear what is the difference between Fig3A and Fig3C ? I did not find any clues in the legends nor in the text to clearly understand that (see last comment below)).
Minor points:
Abstract:
The use of “comprehensive” characterization of CD8 T cells seems largely overstated (5 markers, as listed above) at the time where flow/mass/spectral cytometry allows to analyse much more markers simultaneously.
Figure 1B: the black vs red color in panel A to H of this figure is used for blood vs tumor, so it is confusing in panel B if both black and red correspond to the tumor only (and why is the blood not shown in this panel ?)
Figure 1J: it is unclear what the scale are corresponding for ? For the first line, is it the percentage of “total CD8 T cells” among total CD45+ cells ? (or among total cells of the tumor cell suspension?). And then PD1+, Tim3+ etc.. are related to a color scale going from around -1.5 up to around 2, what is it ? why isn’t it also a percentage among CD8 T cells ?
Fig3A: the author wrote line 180 that there was “2 folds higher” PD1+CD38+Tim3+ cells when comparing both groups, but unless I misunderstand the legend, it is 7.9% vs 25.4% ? so rather >3 folds ?.
And as said above, I it is unclear if these pie charts correspond to the individual data shown in Fig3C, how many patients are in both groups ? 5 vs 22 ? If yes, Fig3C show a median % PD1+CD38+Tim3+ cells in the recurrence group that is <20% (so clearly not 25.4%).
Reviewer 2 Report
Basak and colleagues have studied fresh tissue from primary NMIBC patients with multiplex-flow cytometry and identified suppressed sub-populations of T cells within tumors. The data and analyses are of good quality and the report is well written. However, the number of patients is small limiting the possibility to draw clinically relevant conclusions. The association between T cell fractions and outcome is presented as a treatment effect, but this cannot be concluded from the data. The study is interesting but the conclusion about association to treatment response is overstated. Reviewer: Gottfrid Sjödahl, Lund University, Sweden.
Comments:
1. Throughout, the greek letter gamma and other symbols have been replaced by a strange symbol.
2. Were fresh tumor samples obtained by cold-cup biopsy or by resection with electrocauterization? Include in Materials and methods.
3. Was tumor sampling controlled to include only tissue removed during resection of the superficial tumor (‘bulk’) or was tissue from deeper parts of the bladder wall (i.e. margin) also included?
4. The patients in the study are extremely young for having bladder cancer. It would be interesting to know if this reflects the population at these hospitals or if young patients were selected. If different from the population, the age difference should be mentioned as a limitation of the study.
5. It is a strength that the patients had no prior NMIBCs, because it makes the base-line immune status of the studied tumors much more comparable.
6. Fig1J would be improved if a) Header is included showing the tumor parameters (e.g. stage and grade) b) cases were clustered or ordered by their total CD8+ T cells, and c) individual cell subsets were presented in a logical order, i.e. PD1+, followed by PD1+CD38+, followed by PD1+CD38+Tim3+, etc.
7. Several sub-titles within the results section have been un-formatted. For example ”Intratumoral PD1+CD38+Tim3+ CD8+ T cell subset is positively associated with increased tumor burden in patients with NIMBC” is written in paragraph with the results text, but should probably be a sub-heading.
8. The Figure legends should specify that raw p-values are shown i.e. not corrected for multiple testing. For each significant p-value the text should mention if it would also be significant after applying a suitable multiple testing correction.
9. The observed associations between exhausted T cells fraction and outcome after TUR-BT plus intravesical BCG treatment do not fully support the conclusion that this is treatment related. To explain, Fig 3A shows that patients with a recurrence have a larger fraction of exhausted T cells than patients without recurrence. There are two possibilities: either that would be true even if the same patients would not have recieved BCG, in that case presence of the T cell subset may be a prognostic factor for NMIBC unrelated to the effect of BCG. The other option is that in the (hypothetical) absence of BCG treatment, some patients in the non-recurrence group would be in the recurrence group (the recurrences actually prevented by BCG), and that this transfer of some patients from the left circle diagram to the right one would ‘even out’ the proportional differences observed in the yellow slice. Under this scenario, a BCG-untreated cohort would show no difference in the T cell subset proportions by recurrence-status and the effect seen in the current study would be due to a differential effect of BCG by T cell proportion. The problem is that it is not possible, based on the presented data, to determine which of the two scenarios that is the case. The fact that the EORTC recurrence risk score mirrors the observed recurrence patterns speaks in favor of a BCG unrelated (purely prognostic) role, since it shows that the patients with high PD1+CD38+Tim3+ fraction had high recurrence risk already at baseline. The EORTC recurrence score parameters are derived from multiple largely BCG-untreated cohorts.
To summarize this comment, the data show an association between T cell subset fraction and outcome. However, this difference could exist either regardless of, or due to, the BCG treatment and the current study design does not allow it to be determined. I suggest revising the conclusions throughout and adding a specific discussion stating that that it’s not known to what extent the observed differences in outcome are BCG-related.
Optionally, there’s also an interesting discussion to be had about the following: In a cohort of 27 BCG treated patients, BCG can only have averted a few recurrences, meaning that most patients would be in the same group (recurrence/no recurrence) even if not BCG-treated. These data are simply not enough to determine or even suggest that this is a BCG-predictive biomarker. However, the statement that the T-cell fraction is associated with outcome (recurrence) after BCG treatment is more adequate, because such a statement is open to but does not imply that the outcome differences was due to treatment.
10. Response to BCG cannot be measured directly, so what definition of BCG-response was used? Add the definition in materials and methods. Is Fig3C showing the same thing as Fig3A?
